# Effects of Nordic Walking on Functional Capacity of Women Cohort with Breast Cancer

**Mirela Vuckovic** [1], **Ksenija Bazdaric** [2,*], **Amira Salibasic** [3], **Vlasta Loncar** [4], **Goran Slivsek** [5], **Silvije Segulja** [6] and **Iva Sorta-Bilajac Turina** [4,7]

1    Department of Physiotherapy, Faculty of Health Studies, University of Rijeka, 51000 Rijeka, Croatia; mirela.vuckovic@uniri.hr
2    Department of Basic Medical Sciences, Faculty of Health Studies, University of Rijeka, 51000 Rijeka, Croatia
3    Manus Maris, Trade for Services and Consulting, 10000 Zagreb, Croatia; amira.salibasic@student.uniri.hr
4    Department of Public Health, Teaching Institute of Public Health of the Primorje—Gorski Kotar County, 51000 Rijeka, Croatia; vlasta.loncar@zzjzpgz.hr (V.L.); iva.sorta-bilajac@zzjzpgz.hr (I.S.-B.T.)
5    Center for Applied Bioanthropology, Institute for Anthropological Research, 10000 Zagreb, Croatia; goran.slivsek@xnet.hr
6    Department of Clinical Sciences, Faculty of Health Studies, University of Rijeka, 51000 Rijeka, Croatia; silvije.segulja@fzsri.uniri.hr
7    Department of Environmental Medicine, Faculty of Medicine, University of Rijeka, 51000 Rijeka, Croatia
*    Correspondence: ksenija.bazdaric@fzsri.uniri.hr

**Abstract:** *Background*: Breast cancer is one of the most common tumours and one of the leading causes of death among women in all parts of the world. The aim of this study is to investigate the influence of Nordic walking on the functional capacity of women who have undergone surgery for breast cancer. *Methods*: The study involved a cohort of women who exercised through Nordic walking for 10 weeks (from March to May 2022). The subjects trained with a licenced instructor (INWA method), with two training sessions per week of 70–80 min each. We collected information on pain, arm mobility, hand grip strength, shoulder joint range of motion bilaterally, circumference of both arms, body mass index, physical activity, aerobic capacity, and endurance. *Results*: There were 14 women, median age 63. BMI was significantly lower (28.9/28.1; $p = 0.013$) after training and a difference in shoulder range of motion was better (anteflexion right (142.5/170, $p = 0.002$), retroflexion right (40/60, $p = 0.005$), abduction right (135/180, $p = 0.005$), abduction left (135/180, $p = 0.005$)). There was no difference in right hand strength, while there was a significant difference in left hand strength (19/20, $p = 0.007$). A correlation was found between BMI and the six-minute walk test ($r = -0.70$; $p = 0.005$). *Conclusions*: Considering the multidimensionality of the disease itself and the results of this study, we believe that Nordic walking is a favourable and good choice of physical activity for breast cancer patients.

**Keywords:** breast cancer; hand strength; Nordic walking; physical activity; range of motion; 6MWT

## 1. Introduction

The treatment of breast cancer involves multidisciplinary care and includes surgery, radiotherapy and systemic therapy [1]. Although a multidisciplinary treatment approach improves survival for breast cancer, the treatment methods have both physical and psychological consequences [2]. As a result, the cardiorespiratory system and the musculoskeletal system are affected (predominant weakness and muscle atrophy due to restricted movement). In addition to the psychological and social side effects, one of the common physical side effects of the treatment modalities (as chemotherapy) are pain and lymphoedema of the arm and the associated region, as well as limited mobility of the shoulder joint on the affected side [3].

A lot of evidence suggests positive effects of physical activity on improving the mental health and functional status of women with breast cancer [4–6]. Recent research points

to the importance of early physical activity, which can have a positive impact on early recovery and even reduce the possibility of recurrence [7–9].

Nordic walking (NW) is a form of popular physical activity. Energy consumption is higher when using poles compared to classic walking, while, at the same time, the feeling of fatigue is lower because of using poles [10]. With classic walking, the lower extremities are dominantly active, whereas with NW, the upper body must also be activated because of use of poles. The physical effort is distributed appropriately and evenly across different muscle groups, activating 90% of the skeletal muscles of the whole body. NW is cyclical in nature and helps to condition the body, improve motor skills and, with regular training, also reduce body weight. NW has been associated with benefits in various conditions and diseases in terms of improving the cardiorespiratory system, lipid status, reducing body weight and reducing chronic pain [11–13]. Muscle contractions in the open and closed kinetic chain of the upper extremities create a pumping effect that stimulates the circulation of lymph and blood throughout the arm [10]. According to the recommendations of the WHO, people over the age of 18 should engage in 150 min of moderate-intensity or 75 min of high-intensity physical activity per week [11].

To the best of our knowledge, there are only a very small number of studies that have investigated the effects of NW on functional abilities after breast cancer surgery (PubMed search with keywords "Nordic walking" and "breast cancer" offered 11 results (search performed 14 May 2024)). Therefore, a study was conducted to investigate the effects of NW on the functional abilities of women who had undergone surgery for breast cancer and the effects of BMI on motor tests. Our primary aim was to investigate if NW influences pain, mobility, lymphoedema rates and the impact of NW on functional tests. The secondary aim was to investigate the impact of BMI on the functional capacity of women.

## 2. Materials and Methods

### 2.1. Participants

The study involved women with breast cancer from the city of Rijeka who voluntarily exercised NW (March–May 2022; all participants are members of the Association of Women with Breast Surgery "Nada"—Rijeka, Croatia) and were measured before and after the training intervention. Participants did not engage in NW activities before the study, and they agreed to perform NW within the context of the study. The women were not evaluated prior to surgery; they were selected as members of the Association they joined after the surgery. The main focus of this study was the assessment of intervention's effect.

Originally, 25 women were included in the study, but 14 were eventually included in the analysis. Participants dropped out because they were irregular in their attendance at NW training or were unable to complete the planned ten-week cycle due to complications of the underlying disease.

The criteria for inclusion in the study are women with a diagnosis of breast cancer who were positively assessed by a physician and a physiotherapist in terms of the subjects' participation in physical activity after the surgery and those who agreed to exercise. The exclusion criteria are an incomplete ten-week cycle and inability to exercise due to comorbidities that hinder regular exercise and women who did not have breast cancer surgery.

### 2.2. Nordic Walking

The test subjects trained for 10 weeks with a licenced NW instructor according to the INWA (International Nordic Walking Association) method, with two training sessions per week of 70–80 min each. The technique and learning of NW according to the INWA method are based on three principles: correct walking technique, correct posture and correct pole use. Each training session consisted of a 10 min warm-up, 40–50 min of walking with NW poles, 10 min of strength exercises with dumbbells and elastic bands and 10 min of stretching.

Low-intensity, long-duration cardio exercise was performed with the subjects at 40–60% of the maximum number of heartbeats/minute. During training, the optimum load zone was determined using heart rate measurement, taking age and training goal

into account. Walking took place at a moderate to slightly elevated level of 40–60% of the maximum heart rate predicted for age. The maximum heart rate was determined according to the age formula of 220—age. In this way, a training zone was determined for each woman under the supervision of a licenced instructor. The subjects measured their heart rate every 15 min. In addition to this method, a talk test was also carried out due to the age difference in the group. The talk test is an alternative to the standard measurement methods as it is easy to use and can be used when it is not possible to test people under laboratory conditions [14,15].

*2.3. Measurements*

2.3.1. Parameters

The following parameters were tested before and after the 10-week intervention: body mass index (BMI), limb circumference, hand strength, range of motion, aerobic capacity and endurance, lower limb strength and endurance, and physical activity.

In addition, measurements of the circumference of the arm on the affected side were made by marking both arms, free of clothing, with reference points extending distally from the tip of the middle finger towards the proximal end to the armpit, totalling 60 or 70 cm depending on the length of the arm. The circumference was then measured at each marked point; the values obtained were added together and the difference index between the healthy arm and the affected arm was calculated. The American Physical Therapy Association (APTA) uses circumference as an anthropometric measure to classify lymphoedema. The difference in volume between the affected and healthy side classifies the lymphoedema as: mild (volume difference < 3 cm), moderate lymphoedema (3–5 cm difference) and severe lymphoedema (difference > 5 cm) [16]. Based on the literature, there is no gold standard for measuring lymphoedema. The most commonly used measurement methods are volume calculation and arm circumference measurement [17]. Direct methods such as immersing the limb in water are not practical for clinical use, so many clinicians opt for indirect methods (arm circumference measurement) in their daily work as they are simpler, objective and reliable [18,19]. Circumferential limb measurements were not converted to volume measurements in this study.

Hand strength was measured using a hand dynamometer. The test was performed with the subject sitting with the upper arm next to the body and the elbow bent at a 90-degree angle. The subject repeated the measurement three times with each hand and the mean value was determined.

The range of motion of the shoulder joint was measured using a two-armed goniometer, with one arm fixed and the other movable in a sitting position.

2.3.2. Tests

The six-minute walking test (6 MWT) tests aerobic capacity and endurance. The aim of the test is to cover as many metres as possible in the specified time (6 min) [20].

The 30 s sit to stand test (30 CST) tests the strength and endurance of the lower limbs and has been used in research in tumour patients [21,22]. The average score for women of the same age as in our study is <12 [23].

2.3.3. Questionnaires

We used a basic sociodemographic questionnaire and IPAQ (International Physical Activity Questionnaire)—short form.

Questions regarding pain (in the arm and breast on the affected side) and difficulties in movement of the arm on the affected side (elevation, abduction) in the past week were assessed with a four-point scale (not at all—rarely—often—almost always).

Based on the IPAQ questionnaire, an assessment of energy expenditure is conducted with seven questions, which is expressed by one metabolic equivalent of oxygen (MET). Physical activity is described by four dimensions: frequency, duration, intensity and type of physical activity [24]. All of these variables are constructed in such a way that

they can be presented as separate results for low- (0–600 MET/min/week), medium- (601–3000 MET/min/week) and high- (>3001 MET/min/week) intensity activities or by summing the results [25]. Based on the results of these four variables, the overall level of physical activity was calculated by summing the results. IPAQ has been used in various countries around the world and has shown a high reliability coefficient as a tool for measuring PA levels in numerous international studies. Craig et al. presented the reliability calculated based on studies in twelve countries using the test–retest method over a period of three to seven days [26]. The IPAQ questionnaires showed significant reproducibility of the data (r = between 0.57 and 0.88), demonstrating very good reliability [27].

Data regarding participants' medical condition (surgery type, TNM, stage, therapy) were extracted from participants' medical documentation.

### 2.4. Statistical Analysis

Categorical variables were presented as frequencies and relative frequencies. Due to the small sample size, non-parametric statistics was used and numerical variables were presented as median and interquartile range (IQR). Initial values were compared with other values using the Wilcoxon *t*-test. Correlations between numerical variables were calculated using the Spearman correlation coefficient. *p* values < 0.05 were considered statistically significant. TIBCO 14.0.1.25 Statistics version was used for all presented analyses.

## 3. Results

### 3.1. Patients' Characteristics

The demographic and medical characteristics of the subjects (*n* = 14) are shown in Table 1. Median age of women was 63 (58–71) years. Of the 14 women (with 17 tumours) who took part in the study, 1 woman had cancer on both sides, while 2 women underwent reintervention on the affected side of the upper limb. Detailed information on the participants' medical data is presented in Supplementary S1.

**Table 1.** Participants' demographic and medical characteristics (*n* = 14).

| Characteristic | Variable Category | *n*/Total *n* or C (25–75 Percentile) |
|---|---|---|
| Age | | 63 (58–71) |
| Location of the tumour | Left | 8/14 |
| | Right | 5/14 |
| | Bilateral | 1/14 |
| Dominant hand | Left | 1/14 |
| | Right | 13/14 |
| Type of treatment * | Surgical treatment | 17/14 |
| | Chemotherapy | 11/14 |
| | Radiotherapy | 14/14 |
| | Hormonal therapy | 15/14 |
| | Biological therapy | 3/14 |
| Disease stage * | Stage 0 | 1/14 |
| | Stage I | 8/14 |
| | Stage II | 5/14 |
| | Stage III | 3/14 |
| Number of years since the diagnosis * | 0–5 | 5/14 |
| | 6–10 | 7/14 |
| | 11–15 | 3/14 |
| | 16–20 | 1/14 |
| | 21–25 | 1/14 |

Legend: Stage 0—Carcinoma in situ; Stage I—A tumour confined to the organ of origin; Stage II—A tumour that has spread beyond the organ of origin; Stage III—A tumour that has spread beyond the organ of origin and metastasised to regional lymph nodes. * Applicable to the number of tumours.

*3.2. Difference in Body Mass Index and Arm Circumference before and after Training*

A statistically significant difference in BMI ($p = 0.013$) was found before and after training, such that BMI decreased on average by 0.8 units in 12 women after 10 weeks of training, while there was no statistically significant difference in arm circumference before and after training on either side of the body (Table 2).

**Table 2.** Difference in body mass index and limb circumference before and after training ($n = 14$).

| Variable | C (25–75 Percentile) | p Value | Positive Differences * | Negative Differences | No Change |
|---|---|---|---|---|---|
| BMI 1 BMI 2 | 28.9 (25.7–31.1) 28.1 (25.0–29.1) | 0.013 | 12/14 | 2/14 | 0/14 |
| Right hand 1 Right hand 2 | 171.2 (163.0–179.0) 169.0 (162.5–180.5) | 0.326 | 6/14 | 6/14 | 2/14 |
| Left hand 1 Left hand 2 | 173.0 (164.5–197.0) 173.0 (166.5–187.5) | 0.087 | 9/14 | 4/14 | 1/14 |

Legend: 1 describes the initial measurement and 2 describes the measurement after training; BMI—body mass index; the measurements for both hands are given in cm. * Positive differences are those that are supposed to lead towards better clinical outcome regardless of the mathematical sign (+ or −). Detailed individual differences in all measured variables with positive and negative differences are presented in Supplementary S2 of this article.

Patients whose left side was affected made greater (6/8) progress in reducing the circumference of their arms than the patients whose right side was affected (3/5) (Supplementary S2A,G,H).

*3.3. Arm and Breast Pain and Difficulties in Arm Movement before and after Training*

There were no statistically significant differences in terms of arm pain ($p = 0.205$) and breast pain ($p = 0.724$) before and after training. Difficulties in arm movement were less self-reported after the training [$p = 0.014$; before training (median 2 (25–75 percentile 1–3)) vs. after (median 1 (25–75 percentile 1–3))].

Individual differences in arm and breast pain and difficulties in movement before and after the training are presented in the Supplementary S2B–D. Six patients reported less arm pain, while seven reported no difference in arm pain (Supplementary S2B). For breast pain, six patients reported no change and four reported less pain (Supplementary S2C). As for difficulties in terms of arm mobility, nine patients reported less difficulties and four reported no change (Supplementary S2D). As for participant No. 3, who reported negative differences in all three variables, she underwent chemotherapy 5 days prior to the second measurement.

There was no correlation between arm and breast pain and IPAQ, and hand grip (left and right) before and after the measurement (all $p > 0.05$).

*3.4. The Difference in the Range of Motion of the Shoulder Joint before and after Training*

A difference in the range of motion of the shoulder joint before and after training was found in several areas (Table 3). Range of motion was improved in all areas except for anteflexion ($p = 0.05$) and retroflexion left ($p = 0.051$), where the $p$ value was slightly above the threshold.

Individual differences in measured variables are presented in Supplementary S2 (Tables S2I–N).

**Table 3.** Range of motion.

| Compared Variables in Degrees (°) | C (25–75 Percentile) | *p*-Value | Positive Differences * | Negative Differences | No Change |
|---|---|---|---|---|---|
| Anteflexion right 1<br>Anteflexion right 2 | 142.5(140.0–150.0)<br>170 (160.0–180.0) | 0.002 | 12/14 | 0/14 | 2/14 |
| Anteflexion left 1<br>Anteflexion left 2 | 142.5 (130.0–160.0)<br>160.0 (150.0–180.0) | 0.055 | 10/14 | 1/14 | 3/14 |
| Retroflexion right 1<br>Retroflexion right 2 | 40.0 (40.0–60.0)<br>60.0 (60.0–60.0) | 0.005 | 10/14 | 0/14 | 4/14 |
| Retroflexion left 1<br>Retroflexion left 2 | 55.0 (40.0–60.0)<br>60.0 (60.0–60.0) | 0.051 | 6/14 | 1/14 | 7/14 |
| Abduction right 1<br>Abduction right 2 | 135.0 (110.0–160.0)<br>180.0 (160.0–180.0) | 0.005 | 10/14 | 0/14 | 4/14 |
| Abduction left 1<br>Abduction left 2 | 135.0 (100.0–160.0)<br>180.0 (140.0–180.0) | 0.005 | 10/14 | 0/14 | 4/14 |

Legend: 1 describes the initial measurement and 2 describes the measurement after training. * Positive differences are those that are supposed to lead towards better clinical outcome regardless of the mathematical sign (+ or −). Detailed individual differences in all measured variables with positive and negative differences are presented in Supplementary S2 of this article.

### 3.5. Motor Skills before and after Training

There is no statistically significant difference in hand strength in the right hand, while there is a statistically significant difference in the left hand after 10 weeks of training (Table 4). Fitness and endurance are better after training than at the first measurement regarding the 30 s stand test, 6 min walking test and IPAQ (all $p < 0.005$). Individual differences presented in Table 4 also show the positive results of the training intervention; a detailed report is included in Supplementary S2E,F,O–Q (hand dominance data included).

**Table 4.** Motor skills before and after training (*n* = 14).

| Variables | C (25–75 Percentile) | *p*-Value | Positive Differences * | Negative Differences | No Change |
|---|---|---|---|---|---|
| Hand grip strength right 1<br>Hand grip strength right 2 | 18.0 (14.0–22.0)<br>18.0 (16.0–24.0) | 0.130 | 9/14 | 2/14 | 3/14 |
| Hand grip strength left 1<br>Hand grip strength left 2 | 19.0 (14.0–22.0)<br>20.0 (18.0–28.0) | 0.007 | 9/14 | 0/14 | 5/14 |
| 30sCT 1<br>30sCT 2 | 4.5 (4.0–7.0)<br>6.5 (5.0–8.0) | 0.006 | 13/14 | 1/14 | 0/14 |
| 6MWT1<br>6MWT2 | 408.8 (382.5–450.0)<br>432.5 (390.5–485.5) | 0.044 | 11/14 | 3/14 | 0/14 |
| IPAQ 1<br>IPAQ 2 | 3664.0 (2854.0–4513.0)<br>4479.0 (3862.0–5111.0) | 0.060 | 11/14 | 3/14 | 0/14 |

Legend: 1 describes the initial measurement and 2 describes the measurement after training; 30sCT—30 s sit to stand test; 6MWT—six-minute walking test; IPAQ—International Physical Activity Questionnaires. * Positive differences are those that are supposed to lead towards better clinical outcome regardless of the mathematical sign (+ or −). Detailed individual differences in all measured variables with positive and negative differences are presented in Supplementary S2 of this article.

The influence of BMI on motor skills in relation to the handgrip and aerobic tests was investigated and a strong negative correlation ($r_s = -0.70$) was found between BMI and the 6MWT (Table 5).

**Table 5.** Relationship between body mass index and motor skills after training (n = 14).

| Compared Variables | N | Spearman $r_s$ | *p*-Value |
|---|---|---|---|
| BMI/Grip strength right | 14 | −0.34 | 0.228 |
| BMI/Grip strength left | 14 | −0.02 | 0.933 |
| BMI/6MWT | 14 | −0.70 | 0.005 |
| BMI/30sCT/2 | 14 | 0.09 | 0.753 |

Legend: BMI—body mass index; 30sCT—30 s sit to stand test; 6MWT—six-minute walking test.

In addition, the relationship between BMI and the level of physical activity before training (rs = 0.37; *p* = 0.196) and after training ($r_s$ = −0.22; *p* = 0.445) was not significant.

Since the right hand is dominant in thirteen subjects and the left in only one, we assume that the dominant side had no influence on grip strength. Regardless of dominance, the improvement is the same on both sides (Supplementary S2E,F). In patients whose left side was affected, five of them showed an improvement, and in three subjects, it was the same as before the intervention. In the subjects whose right side was affected, two subjects improved, two remained the same and one worsened. This can be linked to the fact that the dominant side of the body in the female subjects is the right side, which is probably also the stronger side. The patient with bilateral involvement is right-handed, and Table S2F shows that the strength of the hand grip increased more on the left side than on the right.

## 4. Discussion

Although expected, there was no statistically significant difference in arm circumference on either side of the body after the training, but the volumes of both limbs were lower after training (Table 2). Di Blasio et al. [20] found a difference in arm lymphoedema between the group that performed NW only and the group that performed NW and trained according to the ISA method (a programme specifically designed for patients with breast tumours). The reason for this is probably that additional dynamic exercises with and without resistance stimulate muscle activity, which leads to a reduction in oedema and thus arm circumference. In addition to the studies mentioned above, there are studies that describe the positive influence of physical activity and various forms of dynamic exercise on the symptoms of arm lymphoedema [28,29].

BMI was statistically significantly lower after training than before (*p* = 0.013) on average by 0.8 units in almost all women (12/14). BMI and age may represent risk factors for persistent postoperative pain after breast cancer. Higher BMI may also be associated with lower quality of life following breast cancer treatment [30]. Thus, physical activity combined with a reduced calorie diet may represent an important target for preventative strategies in the context of cancer-related pain.

The range of motion of the shoulder joint in our study after NW training is better in most areas than in the initial tests (Table 3). These results are in accordance with some review articles [31,32]. For a normal and controlled range of motion of the shoulder joint, adequate stabilisation of the shoulder girdle is necessary because only with the synergy of all muscles can the movement be performed correctly. The use of poles has a biomechanical effect on the facilitation of both the extensors and the shoulder flexors. In line with our considerations, Hanuszkiewicz et al. [28] conducted a study and found that NW promotes isokinetic endurance of the shoulder girdle muscles better than a general exercise programme in patients with breast tumours. All of this contributes to normal movement. It follows that range of motion is also expected to be better after exercise. Research shows that at all stages of the disease, exercise under professional guidance is not contraindicated. For example, Bruce et al. showed that physiotherapy interventions have a positive effect on shoulder joint range of motion in the early surgical stages of tumour treatment [29]. In addition, Sweeney et al. confirmed that resistance exercises and conditioning exercises influence shoulder function [31].

In our study, left hand grip strength is better after training, while there is no statistically significant improvement in right hand grip strength (Table 4). However, from a clinical point of view, more subjects participated in the study whose breast tumour was located on the left side than those whose breast tumour was located on the right side, while one subject was affected bilaterally. The hand test not only assesses upper limb strength, but also strength and endurance of the whole body [33]. Parra-Soto et al. found in their study that hand grip is also a predictor of some forms of tumours [32]. Bohannon et al. published reference values for healthy women of the same age as our subjects. This value is 23.0 kg for the left hand and 25.9 kg for the right hand [34]. Our subjects measured lower hand strength values on both sides but, after the intervention, there was a statistically significant improvement on the left side (Table 4). Parkinson et al. also found in their study that the hand grip strength of their subjects was weaker than the reference values, indicating the need for organised rehabilitation and physical activity [35]. Ibrahim et al. conducted a study in which they investigated, among other things, the influence of standard exercises on ROM and hand strength and found that there was no statistically significant difference in hand strength after the intervention [36].

In general, fitness and endurance were better after training, as we can see in Table 4. In our study, there was a statistically significant improvement in performance on the 6MWT and 30CST. Eyigor et al. [37] conducted a study with two groups of patients in women who had been cured of breast cancer. One group performed Pilates exercises and the other a home-based programme. There was a statistically significant improvement in the group that practised Pilates compared to the other group. Due to the pathology of the disease itself, lung and heart capacity can be affected, and all this consequently affects the activities of daily life. In our study, the average value determined by this test is much lower (subject sits and stands six times in 30 s) than the average value for the population (sit and stand twelve times in 30 s). A statistically high negative correlation (r = −0.70) with the BMI and 6MWT test was found (Table 5) in our study. A higher BMI is associated with a number of diseases, including reduced mobility.

Considering the multidimensionality of the disease itself and the results of this study, we believe that NW is a favourable and good choice of physical activity for women who have undergone breast surgery. Walking is a very safe form of physical activity, especially when supported with poles. It is scientifically proven how important physical activity is and how important prehabilitation is [38]. In contrast to other forms of exercise, NW is interesting because it also requires being outdoors and in nature, which has an additional proven and already mentioned positive effect and should definitely be the method of choice when selecting exercises.

*Limitations of the Study*

The main methodological limitation of the study that affected the results obtained is the small number of respondents. Due to the nature of the disease, subjects left the study, and it was difficult to predict the final number of participants. In addition, the COVID-19 epidemic was still ongoing, and patients were afraid to participate in research and be in a larger group of people due to their underlying disease. Therefore, additional research with a larger number of subjects is required. A further methodological limitation of this study is that the women enrolled in the Association had surgery in different timeframes, and it was not possible to obtain hospital data from the first clinical measurements.

Furthermore, patients with bilateral breast cancer were included, which can impact lymphoedema measurements if you are not using volume calculations.

The advantage of our study is precisely the observation of a larger number of functional outcomes depending on the intervention—Nordic walking training.

## 5. Conclusions

The range of motion of the shoulder joint is statistically significantly better after training of the anteflexion right, retroflexion right, and abduction on both sides. Left hand

grip strength is better after training, while there is no significant improvement in right hand grip strength. Fitness ability and endurance (hand grip on left side, 30sCT and 6MWT) are better after training than before. After 10 weeks of Nordic walking training, BMI was statistically significantly lower. The association of BMI with three domains (hand strength, 30sCT, 6MWT) showed a statistically significant association only when the 6MW test was performed. There is no statistically significant difference in arm circumference on both sides of the body after training.

**Supplementary Materials:** The following supporting information can be downloaded at: https://www.mdpi.com/article/10.3390/curroncol31060226/s1, Supplementary S1: Participants' medical characteristics; Supplementary S2: Individual differences in measurement outcomes: Table S2A: Body mass index; Table S2B: Arm pain; Table S2C: Breast pain; Table S2D: Difficulties in arm movement; Table S2E: Hand grip right side; Table S2F: Hand grip left side; Table S2G: Arm circumference right; Table S2H: Arm circumference left; Table S2I: ROM_anteflexion_right; Table S2J: ROM_anteflexion_left; Table S2K: ROM_retroflexion_right; Table S2L: ROM_retroflexion_left; Table S2M: ROM_abduction_right; Table S2N: ROM_abduction_left; Table S2O: IPAQ1 and IPAQ2; Table S2P: 6 minute walking test; Table S2Q: 30sCT- 30 second sit to stand test.

**Author Contributions:** M.V. data analysis, manuscript draft, and final approval; A.S., V.L. and G.S. contributed to data collection, interpretation, and final approval; S.S. and K.B. analysis, manuscript draft, and final approval; I.S.-B.T. conception and design of the work, draft reviewing, and final approval. Manuscript was critically reviewed by all the authors; all approved the final version and were accountable for all aspects of the work. All authors have read and agreed to the published version of the manuscript.

**Funding:** This research received no external funding.

**Institutional Review Board Statement:** The study was conducted in accordance with the Declaration of Helsinki, and approved by the Ethics Committee of the Teaching Institute of Public Health of Primorje-Gorski Kotar County (authorisation number 08-820-40/26-22).

**Informed Consent Statement:** Written informed consent was obtained from all subjects involved in the study.

**Data Availability Statement:** The data generated by this research can be obtained from the corresponding author within reasonable requirements.

**Acknowledgments:** We are thankful to the members of the Association of Women with Breast Surgery ("Nada"—Rijeka, Croatia) who took part in this study.

**Conflicts of Interest:** Author Amira Salibasic was employed by the company Manus Maris, Trade for Services and Consulting. The authors declare that there are no conflicts of interest regarding the publication of this article.

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
