# Peer review of "Effects of Nordic Walking on Functional Capacity of Women Cohort with Breast Cancer"

_curroncol, doi:10.3390/curroncol31060226_

Round 1

Reviewer 1 Report

Comments and Suggestions for Authors

The manuscript is well written and results are clearly presented. It could be interesting if authors had some data regarding pain perceived by the patients.

Eventually, they could add this data and perform a correlation between pain an the other parameters, such as hand grip strength or IPAQ.

Author Response

Dear reviewer,

thank you for your review, we have tried to incorporate all your comments. We present our responses below and hope you will be satisfied with the revisions.

  • The manuscript is well written and results are clearly presented. It could be interesting if authors had some data regarding pain perceived by the patients.

Thank you, we have measured breast and arm pain and revised the text accordingly. We have added text in the Methods section (p 4, l 153-155).

  • They could add this data and perform a correlation between pain and the other parameters, such as hand grip strength or IPAQ.

We have added a new section in the Results (p 6, l 209-223) and created a Supplement 2 with the individual differences in measurements, including the pain measurement (Supplement 2, Tables 2 and 3).

Reviewer 2 Report

Comments and Suggestions for Authors

It was a pleasure to review the manuscript “Effects of Nordic Walking on Functional Capacity of Women Cohort with Breast Cancer.”  Vuckovic and colleagues present a prospective study of women undergoing Nordic walking exercise following breast cancer surgery on functional outcomes and breast cancer related lymphedema rates.  Nordic walking differs from classic walking due to the use of pole for uniform upper arm movement, which is thought to provide improved universal exercise as it is distributed evenly across different muscle groups.  The goal of the paper is to demonstrated that Nordic walking improves functional abilities of women undergoing surgery for breast cancer, and lowers breast cancer related lymphedema. 

Comments for the authors:

      1-      Introduction:  First sentence in introduction seems relatively irrelevant to the mission of the manuscript, other than including reduced physical activity and increased body mass index are risk factors.  Consider an introductory sentence commenting on “treatment of breast cancer is a multidisciplinary care which includes surgery, radiotherapy and systemic therapy.  Although a multidisciplinary treatment approach improves survival for breast cancer, the treatment methods have both physical and psychological consequences.”   Line 46-47 states therapeutic procedure, I would state treatment modalities, as chemotherapy has also shown to cause lymphatic dysfunction.  Line 70-71, maybe divide the aims into primary: NW on functional tests, secondary: lymphedema rates and impact of BMI on functional capacity.

      2-      Methods: 

a.       Participants:  Please clarify if the participants performed NW prior to the study, or agreed to perform NW with the study.  Change wording of 14 patients were processed.  “14 patients were included in the analysis.”  Please confirm whether women were evaluated prior to surgery, as baseline functional measurements and lymphedema measurements are essential prior to treatment.

b.       Nordic Walking: Please clarify how you confirmed patients were performing at 40-60% of the maximum number of heart beats/minute.  Were these patients on cardiac monitor during exercise?

c.       Measurements: Greatest concern is in the lymphedema measurements.  There is no mention that measurements were taken prior to surgery, which is critical to lymphedema evaluation.  Secondly there is no mention that volume calculation was made with the circumferential arm measurements.  Furthermore, patients with bilateral breast cancer were included which can impact lymphedema measurements if you are not using volume calculations.  Please confirm if pre-surgical baseline arm evaluations were performed, how frequently post operatively, and if circumferential arm measurements were converted to volume measurements.

d.       Questionnaires:  Please include whether the IPAQ is a validated questionnaire.

e.       Statistical Analysis: Non-parametric based on low sample size, which is a limitation noted by the authors.

       3-      Results:

a.       Patient’s Characteristics (Patient’s is misspelled):  Table 1:  Need information on surgery type: mastectomy vs lumpectomy, and extent of axillary surgery: sentinel lymph node biopsy versus Axillary node dissection.  Would extrapolate more on stage to include specifics on TNM, as number of nodes is important on lymphedema risk.  Would include if chemotherapy included taxane therapy as studies demonstrated effect on lymphatic function. Would include information about extent of radiotherapy: chest wall with or without regional nodal radiation.  Given testing for hand grip strength would also include handedness: right vs. left hand dominant.

b.       Difference in BMI and Arm Circumference:  Please clarify if baseline measurements were performed prior to any surgical intervention.  It appears that you are averaging the results of all patients and comparing the average of all patients before and after NW. It may be more accurate to demonstrated each patient’s pretreatment and post treatment changes, and then look at number of patients that had a change in one direction or another and report on the incidence of change in each study endpoint, or the extent of change in each study endpoint.  But grouping all patients together and averaging can be affected by an outlier.  I.e., if 13 patients had no change and one patient dropped there BMI by 5, then if you just average you will see a statistical decrease in BMI, however 13 patients actually had no change.  I think grouping all the patients results into one is not an accurate evaluation.  Also, lymphedema diagnosis is based on volume measurements and change from baseline, pre-operative baseline is essential.

c.       Difference in the ROM:  Again, all values patients were grouped or averaged.  This really is not an accurate depiction of what is happening in each patient and can be biased by one patient who does extraordinarily well.

d.       Motor Skills before and after training:  Again, all values grouped.  No pre-operative baseline clearly depicted.  Secondly, grip strength can be affected by hand dominance, which was not reported, in addition to impacted on laterality of surgery which needs to be clearly demonstrated.

        4-      Discussion:  Overall, I believe evaluating Nordic Exercise on breast cancer outcomes is an excellent idea.  However, I believe there are serious limitations to the methods and data analysis to make any strong conclusions. 

Overall, I enjoyed reading this single institution prospective study.  I believe it is an excellent study question and has some great potential.  At this time the study design and analysis need to be revised prior to publication.

Comments on the Quality of English Language

Comments in review above.  Minor english language changes.

Author Response

Thank you for your detailed report. We have answered all your queries, added the data and analyses you have requested. We hope you will be satisfied with the revision.

Introduction:  

  • First sentence in introduction seems relatively irrelevant to the mission of the manuscript, other than including reduced physical activity and increased body mass index are risk factors.  Consider an introductory sentence commenting on “treatment of breast cancer is a multidisciplinary care which includes surgery, radiotherapy and systemic therapy.  Although a multidisciplinary treatment approach improves survival for breast cancer, the treatment methods have both physical and psychological consequences.”   

Thank you, the text was revised accordingly ie sentence was deleted (p 1, l 41-46).

  • Line 46-47 states therapeutic procedure, I would state treatment modalities, as chemotherapy has also shown to cause lymphatic dysfunction.  

Thank you, the text was revised accordingly (p 2, l 50-51).

  • Line 70-71, maybe divide the aims into primary: NW on functional tests, secondary: lymphedema rates and impact of BMI on functional capacity.

Thank you, revised accordingly (p 2, l 76-78):. Our primary aim was to investigate if NW influences pain, mobility, lymphoedema rates and the impact of NW on functional tests. The secondary aim was to investigate the im-pact of BMI on functional capacity of women.

 Methods: 

  • Participants:Please clarify if the participants performed NW prior to the study, or agreed to perform NW with the study.  

Thank you, we have explained (p 2, l. 84-87) and inserted: „ Participants did not engage in NW activities before the study, and they agreed to perform NW within the context of the study. The women were not evaluated prior to surgery, they were selected as members of the Association they joined after the operation surgery and the main focus of this study was the assessment of intervention’s effect.“

  • Change wording of 14 patients were processed.  “14 patients were included in the analysis.” 

Thank you, revised accordingly (p 2, l 89-90)..

  • Please confirm whether women were evaluated prior to surgery, as baseline functional measurements and lymphedema measurements are essential prior to treatment.

NO, women were not evaluated prior to the surgery: “The women were not evaluated prior to surgery, because firstly, they were selected as members of the Association they joined after the operation surgery, and secondly, the main focus of this study was the assessment of intervention’s effect.

  • Nordic Walking: Please clarify how you confirmed patients were performing at 40-60% of the maximum number of heart beats/minute.  Were these patients on cardiac monitor during exercise?

Thank you, smart watch was used for measurement. We have explained in detail (p. 3, l 107-116): During training, the optimum load zone was determined using heart rate measurement, taking age and training goal into account. Walking took place at a moderate to slightly elevated level of 40%-60% of the maximum heart rate predicted for age. The maximum heart rate was determined according to the age formula of 220- age. In this way, a training zone was determined for each woman. The subjects measured their heart rate every 15 minutes, all under the supervision of a licenced instructor. In addition to this method, a talk test was also carried out due to the age difference in the group. The talk test is an alternative to the standard measurement methods as it is easy to use and when it is not possible to test people under laboratory conditions [14,15].

  • Measurements: Greatest concern is in the lymphedema measurements.  There is no mention that measurements were taken prior to surgery, which is critical to lymphedema evaluation. 

Thank you, revised accordingly (p.3, l 22-137): In addition, measurements of the circumference of the arm on the affected side were made by marking both arms, free of clothing, with reference points extending distally from the tip of the middle finger towards the proximal end to the armpit, totalling 60 or 70 cm depending on the length of the arm. The circumference was then measured at each marked point, the values obtained were added together and the difference index between the healthy arm and the affected arm was calculated. The American Physical Therapy Association (APTA) uses circumference as an anthropometric measure to classify lymphoedema. The difference in volume between the affected and healthy side classifies the lymphoedema as: mild (volume difference <3cm), moderate lymphedema (3-5cm difference) and severe lymphedema (difference >5cm) [16]. Basede on the literature there is no gold standard for measuring lymphoedema. The most commonly used measurement methods are volume calculation and arm circumference measurement [17]. Direct methods such as immersing the limb in water are not practical for clinical use, so many clinicians opt for indirect methods (arm circumference measurement) in their daily work as they are simpler, objective and reliable [18,19]. Circumferential limb measurements were not converted to volume measurements in this study.

  • Secondly there is no mention that volume calculation was made with the circumferential arm measurements.  

Thank you, please see our previous comment.

  • Furthermore, patients with bilateral breast cancer were included which can impact lymphedema measurements if you are not using volume calculations.  

Thank you, we have noted this in limitations of the study (p 9, l 351-352) although we believe that the direct methods are not practical and indirect methods (that we have used) are also reliable.

  • Please confirm if pre-surgical baseline arm evaluations were performed, how frequently post operatively, and if circumferential arm measurements were converted to volume measurements.

NO, women were not evaluated prior to the surgery, as explained earlier.

  • Questionnaires:  Please include whether the IPAQ is a validated questionnaire.

IPAQ is a well-known validated questionnaire and we have included the explanation in the methods section (p 4, l 163-168): The questionnaire has been used in various countries around the world and has shown a high reliability coefficient as a tool for measuring PA levels in numerous international studies. Craig et al. presented the reliability calculated based on studies in twelve countries using the test-retest method over a period of three to seven days [26]. The IPAQ questionnaires showed significant reproducibility of the data (r = between 0.57 and 0.88), demonstrating very good reliability [27].

  • Statistical Analysis: Non-parametric based on low sample size, which is a limitation noted by the authors.

Thank you.

Results:

  • Patient’s Characteristics (Patient’s is misspelled):  

Thank you, revised accordingly (p 4, l 179).

  • Table 1:  Need information on surgery type: mastectomy vs lumpectomy, and extent of axillary surgery: sentinel lymph node biopsy versus axillary node dissection.  Would extrapolate more on stage to include specifics on TNM, as number of nodes is important on lymphedema risk.  Would include if chemotherapy included Taxane therapy as studies demonstrated effect on lymphatic function. Would include information about extent of radiotherapy: chest wall with or without regional nodal radiation.  

Thank you, the table was revised (p 5, l 187-192)  and all the data we have collected from our participants’ medical documentation are now presented in the Supplement 1.

  • Given testing for hand grip strength would also include handedness: right vs. left hand dominant.

Thank you, data about handedness are included (Table 1 and Supplement 2 - Tables 5 and 6).

  • Difference in BMI and Arm Circumference: Please clarify if baseline measurements were performed prior to any surgical intervention.  It appears that you are averaging the results of all patients and comparing the average of all patients before and after NW. It may be more accurate to demonstrated each patient’s pretreatment and post treatment changes, and then look at number of patients that had a change in one direction or another and report on the incidence of change in each study endpoint, or the extent of change in each study endpoint.  But grouping all patients together and averaging can be affected by an outlier.  I.e., if 13 patients had no change and one patient dropped their BMI by 5, then if you just average you will see a statistical decrease in BMI, however 13 patients actually had no change.  I think grouping all the patients results into one is not an accurate evaluation.  

Thank you, we have now included each patient’s preintervention and postintervention changes, and looked at number of patients that had a change in one direction or another. These changes are presented in Tables 2-4 and Supplement 2. We have created Supplement 2 with individual participants' data and marked positive, negative and no differences. We have included the explanation in the text accordingly.   

  • Also, lymphedema diagnosis is based on volume measurements and change from baseline, pre-operative baseline is essential.

Thank you, we understand the protocol of measurement but we have not conducted this study in the hospital environment nor is that data available to us. Also, although the protocol requires measurement of data before surgery, the collected data is often limited and does not include range of motion, arm circumference and detailed lymphoedema measurement.

  • Difference in the ROM:  Again, all values patients were grouped or averaged.  This really is not an accurate depiction of what is happening in each patient and can be biased by one patient who does extraordinarily well.

Thank you, we have now presented individual changes as described in our previous comment (revised tables 2-4 and supplement 2).

  • Motor Skills before and after training:  Again, all values grouped.  No pre-operative baseline clearly depicted.  Secondly, grip strength can be affected by hand dominance, which was not reported, in addition to impacted on laterality of surgery which needs to be clearly demonstrated

       Thank you, we have now presented individual changes as described previously. Hand dominance and surgery laterality are also reported in Supplement 2.

Discussion: 

  • Overall, I believe evaluating Nordic Exercise on breast cancer outcomes is an excellent idea.  However, I believe there are serious limitations to the methods and data analysis to make any strong conclusions. 

Thank you for your detailed report and constructive criticism. We have added your comments to the limitations of the study.

Round 2

Reviewer 1 Report

Comments and Suggestions for Authors

I think the authors have improved the manuscript as requested.

So it can be accepted in the present form.